# Optimum Method Uploaded Nutrient Solution for Blended Biochar Pellet with Application of Nutrient Releasing Model as Slow Release Fertilizer

**JoungDu Shin [1],\*, SangWon Park [2] and SunIl Lee [1]**

[1]  Department of Climate Change and Agroecology, National Institute of Agricultural Sciences, WanJu Gun 55365, Korea; silee83@rda.go.kr

[2]  Chemical Safety Division, National Institute of Agricultural Sciences, WanJu Gun 55365, Korea; swpark@korea.kr

\*  Correspondence: jdshin1@korea.kr; Tel.: +82-63-238-2494; Fax: +82-63-238-3823

**Abstract:** The nutrient releasing characteristics of a blended biochar pellet comprising a mixture of biochar and pig manure compost ratio (4:6) uploaded with nitrogen (N), phosphorus (P) and potassium (K) nutrient solutions were investigated with the application of a modified Hyperbola model during a 77-day precipitation period. The experiment consisted of five treatments, i.e., the control, as 100% pig manure compost pellet (PMCP), a urea solution made at room temperature (TN), a urea solution heated to 60 °C (HTN), N, P and K solutions made at room temperature (TNPK), and N, P and K solutions heated to 60 °C (HTNPK). The cumulative ammonium nitrogen ($NH_4$-N) in the blended biochar pellets was slow released over the 77 days of precipitation period, but nitrite nitrogen ($NO_3$-N) was rapidly released, i.e., within 15 days of precipitation (Phase I), close behind on a slower release rate within the final precipitation (Phase II). Accumulated phosphate phosphorus ($PO_4$-P) concentrations were not much different, and slowly released until the final precipitation period, while the highest accumulated K amount was 2493.8 mg $L^{-1}$ in the TNPK at 8 days, which then remained at a stage state of K. Accumulated silicon dioxide ($SiO_2$) concentrations abruptly increased until 20 days of precipitation, regardless of treatments. For the application of the releasing model for nutrient releasing characteristics, the estimations of accumulated $NH_4$-N, $NO_3$-N, $PO_4$-P, K and $SiO_2$ in all the treatments were significantly ($p < 0.01$) fitted with a modified Hyperbola model. These findings indicate that blended biochar pellets can be used as a slow release fertilizer for agricultural practices.

**Keywords:** blended biochar pellet; modified Hyperbola; nutrient release; slow release fertilizer

## 1. Introduction

Biomass from the agricultural sector is composed of agricultural wastes such as crop residues and fringing trees. Researchers have shown that agro-biomass derived carbon materials such as biochar are one (1) option for carbon sequestration as well as the reduction of greenhouse gas emissions. It is estimated that Korea produces 50 million tons of organic waste from agriculture every year [1]. Thus, to address concerns about climate change, carbon sequestration using biomass conversion technology should be prioritized, since it has already become feasible in Korea. Biochar consists of multi-porous and carbonaceous material obtained from biomass conversion technology under limited oxygen. It consists of non-degradable carbon with double bonds and an aromatic ring that cannot be broken down by microbial degradation [2]; it could be applied for several purposes [3]. However, 30% of biochar is lost due to wind, while 25% is lost due to runoff water during spreading in cultivation areas [4].

It has been suggested that Biochar's effects come from a plant's adsorption and retention abilities vis-à-vis the available nutrients [5]. Nitrogen fertilizers have been chemically transformed through

mineralization, ammonification, nitrification, de-nitrification, and immobilization in soil [6,7]. Biochar from rice hull through biomass conversion technology was reported to have 0.498 mg g$^{-1}$ of maximum absorption capacity of $NH_4$-N, which can prevent nitrification [8]. Biochar from holm oak tree increased $NH_4$-N adsorption in sandy Acrisol, but had no effect on $NO_3$-N adsorption in the column experiment [9]. However, the maximum adsorption of $NH_4$-N in the biochar pellet is 2.94 mg·g$^{-1}$, and lettuce yield was enhanced by approximately 13% relative to the control [10]. When biochar pellet is applied to the soil, the application of chemical fertilizers is still needed to enhance crop productivity. Previous studies on the characterization of the $NH_4$-N adsorption capacity of biochar pellets recommends the mixing of biochar with compost containing nutrients before the soil application [10,11]. Phosphorous (P) is essential for maintaining profitable crop production [12,13]. However, it might cause eutrophication in lakes and ponds, thereby destroying the ecosystem [14]. In recent years, attention for biochar has increased because of its absorption abilities [15,16]. Manure-derived biochar increased the maximum uptake of nitrogen (N), phosphorus (P) and potassium (K) by ryegrass by 66.4%, 161% and 210%, respectively [17]. Biochar from tree prunings enhanced P use efficiency from organic P fertilizer more than the chemical fertilizers for both corn and wheat crops [18]. Silicon (Si), on the other hand, is important to the strength of cell walls and defense against diseases, as well as to enhancing the uptake of other nutrients, and increasing tolerance to drought, salts, and extreme temperature stresses [19–21]. Silicon plays an important role in improving the rice yields. It is a potent stimulator of photosynthesis [22].

Biochar is not suitable for crop cultivation in a practical view because it does not contain sufficient amounts of plant nutrients for crop production. Biochar pellets are an efficient way to decrease field handling costs and to significantly reduce biochar loss during soil application [23]. For soil incorporation, poultry litter was mixed, pelletized and slowly pyrolyzed to produce biochar pellets [24]. However, little information is available on blended biochar pellets as a slow release fertilizer, which is required to allow nutrients to flow slowly from the soil during the cultivation period. It supplies most nutrients into the crop without leaching and denitrification losses [25] to increase profits and minimize environmental destruction [26].

The predictions of the used model confirmed that Michaelis–Meten-type kinetics is probably the most dominant mechanism for the leaching of heavy metals from cement-based waste forms. Furthermore, Michaelis–Menten kinetics have been used to explore nitrogen deposition and climate change in laboratory experiments [27]. For the optimization of the blended biochar pellets, it was described that the release of plant nutrient amounts demonstrated a modified hyperbola model [10].

It is hypothesized that (1) the blended biochar pellet with different nutrient uploaded methods have different means of releasing nutrients, and (2) an optimal nutrient uploaded method can be selected using a modified hyperbola model.

Therefore, this experiment aimed to select an optimum nutrient uploaded method with plant nutrient solutions for blended biochar pellets using a Hyperbola model for agricultural practices.

## 2. Materials and Methods

### 2.1. Biochar Pellet Productions

Biochar (Go-Chang, JenBok) from rice hull and pig manure compost (NOUSBO Co., Suwon, Korea) were purchased from a local farming cooperative union. The biochar produced from the pyrolysis system is described in detail in Shin's previous publication [10]. The physicochemical properties of biochar and pig manure compost used in this experiment are presented in Table 1. The pH of biochar and pig manure compost were 9.8 and 8.8, and their total carbon contents were 566.3 and 288.9 g kg$^{-1}$, respectively.

**Table 1.** Physicochemical properties of biochar and pig compost used [1].

| | pH | EC (dS m$^{-1}$) | TC | TOC | TIC | TN |
|---|---|---|---|---|---|---|
| | | | ------------------- g kg$^{-1}$ ------------------- | | | |
| Biochar | 9.78 (1:20) | 16.53 | 575.5 | 533.0 | 42.5 | 2.0 |
| Pig manure compost | 8.77 (1:5) | 3.40 | 288.8 | 258.6 | 30.2 | 29.1 |

[1] TC; Total carbon, TOC; Total organic carbon, TIC; Total inorganic carbon, and TN; Total nitrogen. The values are represented mean of triplicates samples.

The chemical properties of different blended biochar pellets uploaded with plant nutrient solutions before use in experiments are presented in Table 2. Biochar pellets uploaded with plant nutrient solutions heated to 60 °C had greater than 1.43–1.80% more total nitrogen (T-N) than those at room temperatures; however, their K concentrations decreased.

**Table 2.** The chemical properties of different biochar pellets before used experiments.

| Treatments * | T-N | T-P | K | SiO$_2$ |
|---|---|---|---|---|
| | ------------------------ g kg$^{-1}$ ------------------- | | | |
| TN | 84.0 | 35.4 | 13.5 | 119.5 |
| HTN | 102.0 | 29.5 | 11.8 | 125.5 |
| TNPK | 75.2 | 32.8 | 57.2 | 108.6 |
| HTNPK | 89.5 | 35.6 | 39.0 | 96.9 |
| PMCP | 29.1 | 79.4 | 20.8 | 67.2 |

*TN, uploaded with urea solution at room temperature, HTN; urea solution heated at 60 °C, TNPK; N, P and K solutions at room temperature, HTNPK; N, P and K solutions heated at 60 °C and PMCP; pig manure compost pellets as control. The values are represented as means of triplicate samples.

Different blended biochar pellets were produced by a combination of biochar and pig manure compost (4: 6 ratio, w/w) uploaded with N, P and K plant nutrient solutions. The blended biochar pellets were differently uploaded using a urea solution at room temperature (TN), a urea solution heated to 60 °C (HTN), N, P and K solutions at room temperature (TNPK), and finally, N, P and K solutions heated to 60 °C (HTNPK) to increase solubility. The combination material (2.5 kg total weight) was thoroughly mixed at a 4:6 ratio of biochar and pig manure compost using an agitator (SungChang Co., KyungGi, Korea) for five minutes. The agitator was run continuously while spraying with 1 L of each nutrient solution for ten minutes. The blended biochar pellet (Ø 0.51 cm × 0.78 cm) were made using a machine (7.5 KW, 10 HP, KumKang Engineering Pellet Mill Co., DaeGu, Korea) and by pouring the combination of materials. A processing diagram of the biochar pellet has already been described by Shin et al. [10].

## 2.2. Batch Column Experiment of Nutrient Precipitation and Chemical Analysis

The experiments consisted of five treatments, i.e., PMCP as control, TN, HTN, TNPK, and HTNPK. Into each glass column was placed a filter and 5 g of different blended biochar pellets. Then, 50 mL deionized water was added. Successive precipitations were performed for nutrient extraction, and the 50mL of deionized water was replaced over a period of 77 days. The collected water was filtered using Whatman No. 2 filter paper. It was then analyzed for $NH_4$-N and $NO_3$-N, and the rest of the water samples were stored in a refrigerator until the analysis of $PO_4$-P, K, and $SiO_2$ were undertaken using a UV spectrophotometer (ST-Ammonium, C-Mac, Korea). Total carbon (TC) and total organic carbon (TOC) contents were measured using a TOC analyzer (Elementar Vario EL II, Hanau, Germany). Total P, K, and Si in different types of the blended biochar pellets were analyzed using inductively-coupled plasma atomic emission spectrometry (ICP-AES, IntegraXL, GBC LTd., Braeside, Australia) after digesting the samples with nitric and hydrochloric acids.

### 2.3. Releasing Nutrient Model

The application of a modified Hyperbola model for selecting an optimum mixing ratio of pig manure compost and biochar to make blended biochar pellets based on a nutrient releasing model [10]. In this study, the estimation of the accumulated releasing nutrient amounts in the treatments fitted well with the modified hyperbola model. Therefore, a modified Hyperbola model from the Michaelis-Menten equation was applied to select an optimal nutrient uploading method for the blended biochar pellets.

$$Y = Amax\ [t]/(t1/2_{(Amax)} + [t]) \tag{1}$$

where Y is the accumulated concentration (mg $L^{-1}$); Amax: the maximum accumulated concentration (mg $L^{-1}$); $t1/2_{(Amax)}$: the required time to reach 1/2 Amax; and t the precipitation periods (in days).

### 2.4. Statistical Analysis

The validity of each parameter for a modified Hyperbola model was assured for a normal distribution using Shapiro–Wilk test ($p < 0.05$). The nutrient releasing model for each treatment was accessed by analysis of data using SigmaPlot 12 (Systat Software, Inc., San Jose, CA, USA). The model based on the correlation coefficient values ($R^2$) was estimated. The statistical analyses for the total water-soluble amounts of $NH_4$-N, $NO_3$-N, $PO_4$-P, K, and $SiO_2$ were performed using one-way ANOVA with 5 levels, using SAS version 9.0 (SAS Institute, Carry, NC, USA). After deciding on the significant differences ($p < 0.0001$) among treatment means with analyses of variances (ANOVA), Duncan's multiple range tests were performed for each parameter throughout the whole precipitation period. The means of variables among treatments were compared with parameters using the above equation, according to $p$-value $< 0.0001$, after ANOVA analysis.

### 3. Results

It appeared that the highest accumulated $NH_4$-N concentration was 371.6 mg $L^{-1}$ in the HNPK, but it was almost the same between the HTN and the TNPK during the final precipitation period (Figure 1). The highest accumulated $NO_3$-N concentration was 101.0 mg $L^{-1}$ in the HTN throughout the experimental period, but results did not differ significantly ($p > 0.05$) among the other treatments within Phase I. The order of highest $NO_3$-N release in phase II was HTN>TNPK>HTNPK>TN>PMCP.

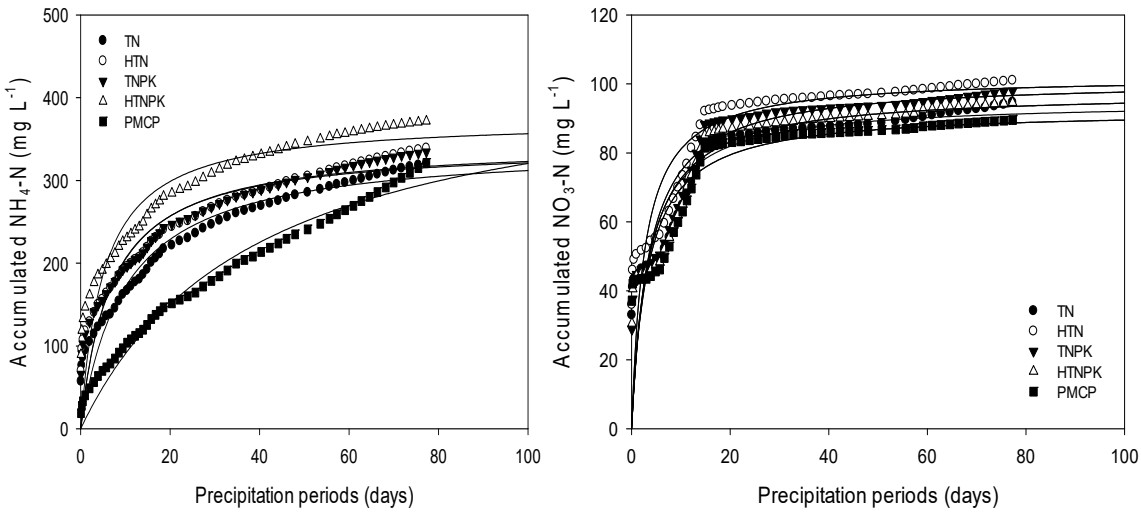

**Figure 1.** Releasing curves of $NH_4$-N and $NO_3$-N from different blended biochar pellets uploaded with plant nutrient solutions. The results are the mean of triplicate samples. Error bars indicate the standard deviation.

The required times of 1/2 maximum accumulated $NH_4$-N and $NO_3$-N levels were taken during 6.7 and 3.5 days in the TNPK, respectively (Tables 3 and 4). The model was significantly fitted ($p < 0.01$) and correlated with the $R^2$ values between the observed and estimated values.

**Table 3.** Estimation model for $NH_4$-N release from different types of the blended biochar pellet.

| Treatments * | Model Parameters | | | | Analysis of Variance | | $R^2$ |
|---|---|---|---|---|---|---|---|
| | Amax | *p*-Values | $t_{1/2(Amax)}$ | *p*-Values | F | *p*-Values | |
| TN | 343.1 | <0.0001 | 10.0 | <0.0001 | 566.7 | <0.0001 | 0.916 |
| HTN | 345.2 | <0.0001 | 6.9 | <0.0001 | 348.1 | <0.0001 | 0.870 |
| TNPK | 342.7 | <0.0001 | 6.7 | <0.0001 | 405.2 | <0.0001 | 0.886 |
| HTNPK | 374.6 | <0.0001 | 5.1 | <0.0001 | 266.0 | <0.0001 | 0.837 |
| PMCP | 446.9 | <0.0001 | 39.4 | <0.0001 | 1962.5 | <0.0001 | 0.974 |

*TN, uploaded with urea solution at room temperature, HTN; urea solution heated to 60 °C, TNPK; N, P and K solutions at room temperature, HTNPK; N, P and K solutions heated to 60 °C and PMCP; pig manure compost pellet as control. Means values indicate significant differences ($p < 0.001$) among treatments (ANOVA).

**Table 4.** Estimation model for $NO_3$-N release from different types of the blended biochar pellet.

| Treatments * | Model Parameters | | | | Analysis of Variance | | $R^2$ |
|---|---|---|---|---|---|---|---|
| | Amax | *p*-Values | $t_{1/2(Amax)}$ | *p*-Values | F | *p*-Values | |
| TN | 95.0 | <0.0001 | 3.2 | <0.0001 | 136.5 | <0.0001 | 0.724 |
| HTN | 102.1 | <0.0001 | 2.6 | <0.0001 | 127.5 | <0.0001 | 0.710 |
| TNPK | 101.1 | <0.0001 | 3.5 | <0.0001 | 202.0 | <0.0001 | 0.795 |
| HTNPK | 97.7 | <0.0001 | 3.5 | <0.0001 | 161.5 | <0.0001 | 0.756 |
| PMCP | 92.5 | <0.0001 | 3.3 | <0.0001 | 111.5 | <0.0001 | 0.682 |

*TN, uploaded with urea solution at room temperature, HTN; urea solution heated to 60 °C, TNPK; N, P and K solutions at room temperature, HTNPK; N, P and K solutions heated to 60 °C and PMCP; control as pig manure compost pellet. Means values indicate significant differences ($p < 0.001$) among treatments (ANOVA).

The accumulated $PO_4$-P concentration in precipitation water from the different treatments is shown in Figure 2. The lowest accumulated $PO_4$-P concentration in the PMCP was 423.2 mg $L^{-1}$ from 10 days through 55 days of the precipitation (phase II), with a rapid increase occurring between 55 and 77 days of precipitation.

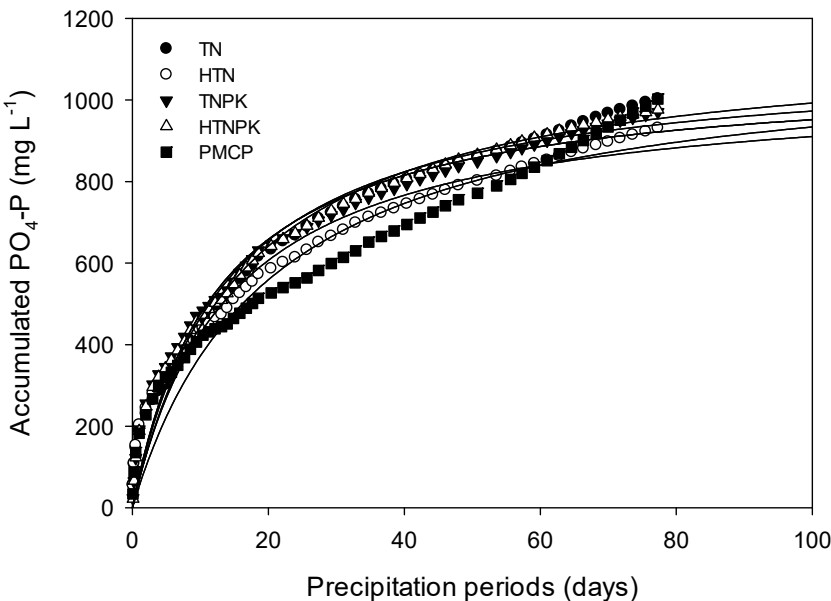

**Figure 2.** Releasing curves of $PO_4$-P from the blended biochar pellet uploaded with different plant nutrient solutions. Results are the mean of triplicate samples. Error bars indicate standard deviation.

It was observed that the required times of 1/2 the maximum amount of accumulated PO$_4$-P were taken from 12.6 to 15.9 days, regardless of the blended biochar pellet uploaded with different nutrient solutions, but that they were taken from 20.1 days in the PMCP (Table 5).

**Table 5.** Estimation model for PO$_4$-P release from different types of the blended biochar pellet.

| Treatments * | Model Parameters | | | | Analysis of Variance | | R$^2$ |
|---|---|---|---|---|---|---|---|
| | Amax | *p*-Values | t$_{1/2(Amax)}$ | *p*-Values | F | *p*-Values | |
| TN | 1150.0 | <0.0001 | 15.9 | <0.0001 | 2454.8 | <0.0001 | 0.979 |
| HTN | 1040.6 | <0.0001 | 14.3 | <0.0001 | 1436.4 | <0.0001 | 0.965 |
| TNPK | 1071.1 | <0.0001 | 12.6 | <0.0001 | 2204.4 | <0.0001 | 0.977 |
| HTNPK | 1106.7 | <0.0001 | 13.8 | <0.0001 | 2496.4 | <0.0001 | 0.980 |
| PMCP | 1122.0 | <0.0001 | 20.1 | <0.0001 | 750.9 | <0.0001 | 0.935 |

*TN, uploaded with urea solution at room temperature, HTN; urea solution heated to 60 °C, TNPK; N, P and K solutions at room temperature, HTNPK; N, P and K solutions heated to 60 °C and PMCP; pig manure compost pellet as control. Means values indicate significant differences ($p < 0.001$) among treatments (ANOVA).

The accumulated K concentrations rapidly increased until 7 days of precipitation, and then remained in a steady state through 30 days of precipitation, regardless of the treatment (Figure 3). The study showed that the highest accumulated K concentration was 2680.5 mg L$^{-1}$ in the TNPK, and that the lowest was in the TN and HTN which unloaded with plant nutrient solutions, even though those mixed with 60% of pig manure compost were 1729.0 and 1754.5 mg L$^{-1}$, respectively.

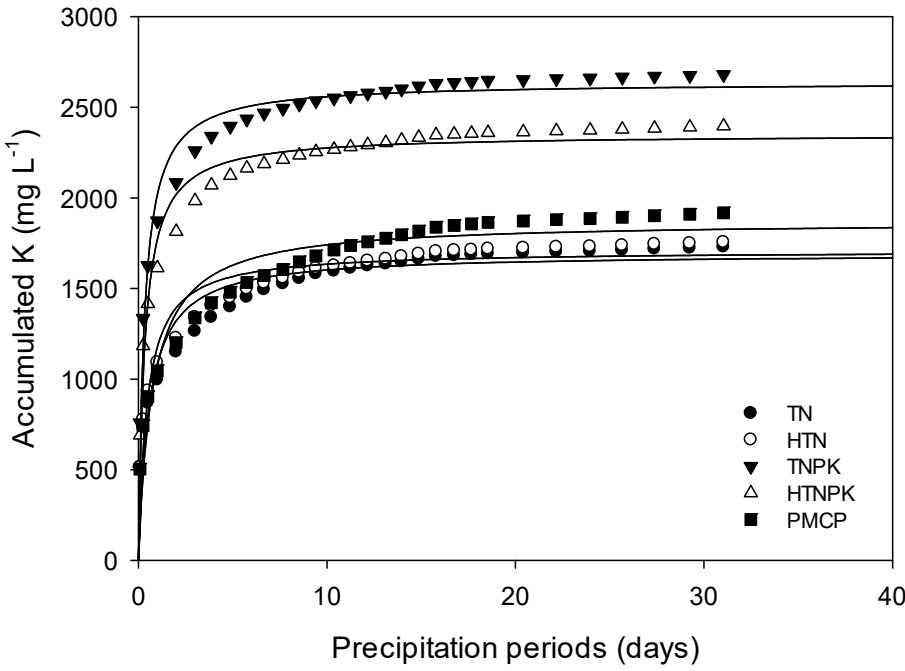

**Figure 3.** Releasing curves of K from the blended biochar pellet uploaded with plant nutrient solutions. The results are the mean of triplicate samples and error bars indicate standard deviation.

The estimation model for K releases from different blended biochar pellets calculated using a modified hyperbola equation is presented in Table 6. The required times of 1/2 maximum accumulated K amount were taken less than 0.6 days in all the treatments because of a high water-soluble capacity (Table 6).

**Table 6.** Estimation model for K release from different types of the blended biochar pellet.

| Treatments * | Model Parameters | | | | Analysis of Variance | | $R^2$ |
|---|---|---|---|---|---|---|---|
| | Amax | *p*-Values | $t_{1/2(Amax)}$ | *p*-Values | F | *p*-Values | |
| TN | 1692.3 | <0.0001 | 0.6 | <0.0001 | 266.5 | <0.0001 | 0.905 |
| HTN | 1708.6 | <0.0001 | 0.5 | <0.0001 | 323.7 | <0.0001 | 0.920 |
| TNPK | 2637.4 | <0.0001 | 0.3 | <0.0001 | 759.0 | <0.0001 | 0.964 |
| HTNPK | 2350.4 | <0.0001 | 0.3 | <0.0001 | 576.4 | <0.0001 | 0.954 |
| PMCP | 1870.0 | <0.0001 | 0.7 | <0.0001 | 258.1 | <0.0001 | 0.902 |

*TN, uploaded with urea solution at room temperature, HTN; urea solution heated to 60 °C, TNPK; N, P and K solutions at room temperature, HTNPK; N, P and K solutions heated to 60 °C and PMCP; control as pig manure compost pellet. Means values indicate significant differences ($p < 0.001$) among treatments (ANOVA).

It appeared that the highest accumulated $SiO_2$ concentration was 2935.7 mg $L^{-1}$ in the TN, but while lowest in the PMCP was 1599.6 mg $L^{-1}$ in the final precipitation periods. The lowest in the PMCP was continuously observed, compared to the other treatments from 6.7 days through to the end of precipitation period (Figure 4).

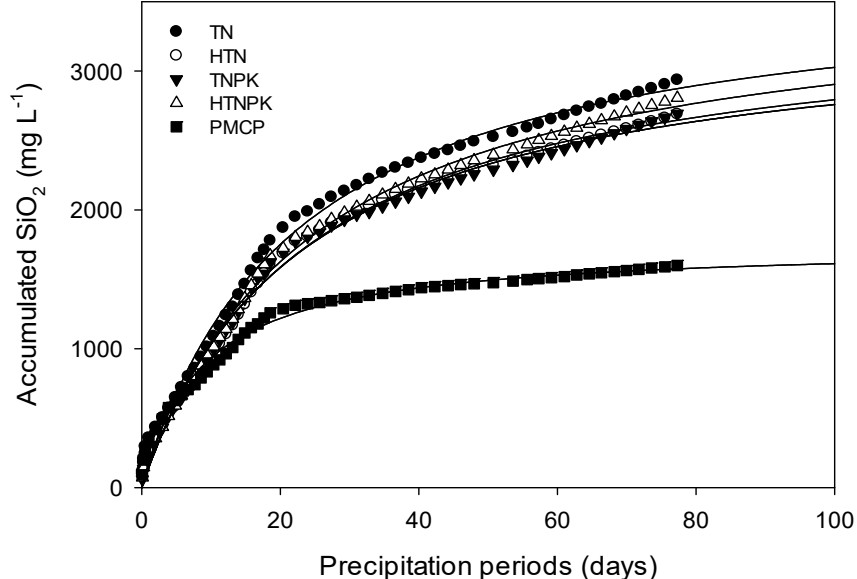

**Figure 4.** Releasing curves of $SiO_2$ from the blended biochar pellet uploaded with different plant nutrient solutions. The results are mean of triplicate samples. Error bars indicate standard deviation.

The estimation model for $SiO_2$ releases from different blended biochar pellets is presented in Table 7. The required times of 1/2 the maximum amount of accumulated $SiO_2$ were taken from 22.3 to 24.4 days, regardless of the blended biochar pellet uploaded with plant nutrient solutions, but were taken at 8.8 days in the PMCP.

**Table 7.** Estimation model for $SiO_2$ release from different types of the blended biochar pellet.

| Treatments * | Model Parameters | | | | Analysis of Variance | | $R^2$ |
|---|---|---|---|---|---|---|---|
| | $A_{max}$ | *p*-Values | $t_{1/2(Amax)}$ | *p*-Values | F | *p*-Values | |
| TN | 3700.1 | <0.0001 | 22.3 | <0.0001 | 7328.6 | <0.0001 | 0.993 |
| HTN | 3448.0 | <0.0001 | 23.4 | <0.0001 | 8354.6 | <0.0001 | 0.994 |
| TNPK | 3386.0 | <0.0001 | 22.7 | <0.0001 | 9683.2 | <0.0001 | 0.995 |
| HTNPK | 3612.0 | <0.0001 | 24.4 | <0.0001 | 11815.5 | <0.0001 | 0.996 |
| PMCP | 1754.9 | <0.0001 | 8.8 | <0.0001 | 3128.4 | <0.0001 | 0.984 |

*TN, uploaded with urea solution at room temperature, HTN; urea solution heated to 60 °C, TNPK; N, P and K solutions at room temperature, HTNPK; N, P and K solutions heated to 60 °C and PMCP; control as pig manure compost pellet. Means values indicate significant differences ($p < 0.001$) among treatments (ANOVA).

The total water-soluble amounts of $NH_4$-N, $NO_3$-N, $PO_4$-P and $SiO_2$ were significantly different ($p < 0.01$) among TN, HTN TNPK and HTNPK, but all soluble amounts in the PMCP were significantly lower than those of the other treatment throughout the precipitation period (Table 8).

**Table 8.** Comparisons of the total water-soluble accumulated amounts of $NH_4$-N, $NO_3$-N, $PO_4$-P, K, and $SiO_2$ for different treatments during the precipitation periods.

| Treatments * | Total Water-Soluble Amounts (mg) | | | | |
|---|---|---|---|---|---|
| | $NH_4$-N | $NO_3$-N | $PO_4$-P | K | $SiO_2$ |
| TN | 2935.7 a | 94.7 ab | 984.8 | 1729.1 d | 2935.7 a |
| HTN | 2689.1 a | 101.1 a | 931.4 | 1754.5 d | 2689.1 a |
| TNPK | 2693.1 a | 97.9 a | 920.1 | 2680.5 a | 2693.1 a |
| HTNPK | 2806.4 a | 94.5 ab | 975.8 | 2395.5 b | 2806.4 a |
| PMCP | 131.0 b | 89.5 b | 995.6 | 1971.4 c | 1599.6 b |
| F | 53.40 | 5.64 | 13.42 | 142.62 | 53.40 |
| *p*-values | 0.0003 | 0.0427 | 0.0706 | <0.0001 | 0.0003 |

*TN, uploaded with urea solution at room temperature, HTN; urea solution heated to 60 °C, TNPK; N, P and K solutions at room temperature, HTNPK; N, P and K solutions heated to 60 °C and PMCP; pig manure compost pellet as the control. Means values followed by different letters indicate significant differences ($p < 0.0001$) among treatments with $NH_4$-N, $PO_4$-P, K, and $SiO_2$ (ANOVA and subsequent Duncan Multiple Range Test).

## 4. Discussions

### 4.1. Nitrogen Release from the Blended Biochar Pellet

Nitrogen fertilizers in soil usually undergo nitrogen transformation processes such as mineralization, nitrification, de-nitrification, and immobilization [6,7]. Urea is used as a nitrogen fertilizer because of fast N uptake by crop, although only 40% is absorbed by the plant, while 60% is lost in various ways [28], the great part of which, i.e., 26.5% to 29.4%, is lost through evaporation, thereby contributing to greenhouse gases. Therefore, a controlled release N fertilizer is the best way to minimize $N_2O$ emissions from soil [29]. Shin et al. [10] observed that $NH_4$-N was adsorbed rapidly, with a combination rate (9:1, biochar: pig manure compost) of the blended biochar pellets in both the pseudo first and second order kinetics. It was further observed that the more biochar in the blended biochar pellet, the greater the adsorption of $NH_4$-N.

With respect to the estimation values from a modified hyperbola model, it appeared that half of the nitrogen was released within 10 days of precipitation, while though the rest was slowly released through end of experiment (Tables 3 and 4). However, it was observed that the cumulative $NH_4$-N in the blended biochar pellets was slowly released over the course of the 77 days of precipitation, but that the $NO_3$-N was rapidly released within 15 days of precipitation (Phase I), closely followed by a slower release rate in the final precipitation (Phase II), regardless of the treatment (Figure 1). Thus, this study demonstrated that blended biochar pellets can work as an ideal slow release fertilizer to create

an eco-friendly environment through the reduction of non-point source pollution, and that they may be included in a carbon trading mechanism.

Different patterns for the estimation model were observed in the accumulated $NH_4$-N and $NO_3$-N releasing model. The estimated releasing model was fit wee ($p < 0.01$) with all treatments (Tables 3 and 4).

### 4.2. $PO_4$-P, K, and $SiO_2$ Releases from the Blended Biochar Pellet

P, K and $SiO_2$ are essential elements for crop cultivation; these applications are needed to increase crop yields [12,13]. Phosphate fertilizers are applied to soil every year to increase soil fertility. Annually, current consumption of rock phosphorous as a fertilizer is more than one million tons [30]. The excessive flow of phosphorous in lakes and ponds from croplands is a major cause of eutrophication, which disturbs the ecosystem [14]. Busted eutrophication not only affects aquatic ecosystems, but also indirectly interferes with economic development [31]. Kim et al. [32] reported that two apparent phases showed curves, a greater nutrient release rate within one day (Phase I) providing nutrients to crop growth for short times, followed by a slower nutrient release within 18 days (Phase II) providing nutrients to crops for longer periods. However, the accumulated $PO_4$-P concentrations were rapidly increased up to 77 days of the precipitation period, but K was almost in a steady state after 20 days of precipitation, regardless of treatments. The releasing patterns of $PO_4$-P and K did not agree with Kim's experimental results. On the other hand, this implies that the blended biochar pellets used in this study released plant nutrients more slowly than those used in Kim's study.

Releasing $PO_4$-P concentrations in the TNPK and HTNPK were similar compared with those of TN and HTN at 20 days' precipitation (Phase I), even if not uploaded with phosphorous fertilizer. The maximum accumulated concentrations were not significantly different ($p > 0.05$) among all treatments with different uploaded methods of plant nutrient solutions, even if not uploaded with P; this might be attributed to the P content in pig manure compost. The curves were derived from the estimation model calculated by a modified hyperbola equation using the correlation coefficient value ($R^2$). The estimation models for $PO_4$-P releases from different blended biochar pellets indicated that estimation values were well fitted ($p < 0.01$) and that they correlated with the observed values for $PO_4$-P releases, regardless of which blended biochar pellets were used (Table 5).

Furthermore, the K releasing pattern was similar to the results of this research, except regarding the blended biochar pellet mixed with pig manure compost and uploaded with plant nutrient solutions, which had 6 and 12 days' longer releasing periods in Phases I and II, respectively. This might be due to the different mixing materials, i.e., pig manure compost instead of lignin. The accumulated K concentrations in the TN and HTN did not differ significantly ($p > 0.05$) from the PMCP. The estimation values from a modified hyperbola model were significantly correlated ($p < 0.01$) with the observed values for K releases, regardless of which blended biochar pellets were used.

Silicon (Si) in the soil existed in an unavailable form for plant uptake. Si in crop residues is a useful form relative to that from Si fertilizer. This recycling of Si form can occur on crop land after the decomposition of crop residues. However, Si in crop residues is often removed during harvesting [33]. Also, Si is generally available as uncharged monosilicic acid ($H_4SiO_4$) [20]. Accumulated $SiO_2$ concentrations in different treatments rapidly increased up to 18 days of precipitation (Phase I), and then gradually increased up to 77 days (Phase II), regardless of the treatment (Figure 4). The accumulated $SiO_2$ concentration in precipitation water did not significantly differ ($p > 0.05$) among the HTN, TNPK, and HTNPK. The K and $SiO_2$ releasing patterns showed a similar trend [32], but the $PO_4$-P in this study was released rapidly at the end of the precipitation period. The estimation values calculated from a modified hyperbola equation were well fitted ($p < 0.01$) and correlated with the observed values for $SiO_2$ release, regardless of which blended biochar pellets were used.

Regarding the total water soluble amounts of plant nutrients, no differences were observed with the uploaded methods for blended biochar pellets, especially for $PO_4$-P. However, optimal slow release fertilizer could depend on 1/2 releasing periods, but was still considered to be releasing for the rest of

the precipitation periods, because of the additional fertilizer application method for rice cultivation. It was considered that the best uploaded method of plant nutrients might be the TNPK for the blended biochar pellet.

It was also shown that the experimental results were in agreement with Shin and Park's experimental data [10]. Furthermore, Shin's principal law indicated that pig manure compost pellets released nutrients more slowly than pig manure compost. Therefore, this theory was applied for the blended biochar pellets uploaded with plant nutrient solutions as a slow release fertilizer having an adsorption capacity for plant nutrients.

## 5. Conclusions

This experiment sought to select an optimum nutrient uploading method for blended biochar pellets using a modified hyperbola model based on their nutrient releasing characteristics. The results of this experiment concluded that blended biochar pellets slowly release nutrients over a 77 day precipitation period, and that they create an eco-friendly environment by reducing non-point pollutants. It appeared that the optimal uploaded method was the TNPK based on the nutrient releasing characteristics with a modified hyperbola model. Furthermore, it was observed that a modified hyperbola model fitted well ($p < 0.01$) in all treatments. Furthermore, dose responses for the blended biochar pellet application in the cropland during crop cultivation should be measured in the future.

Overall, these results showed that blended biochar pellets can be used as a slow release fertilizer for agriculture.

**Author Contributions:** Conceptualization, funding acquisition and writing—original draft preparation, J.S.; visualization and software, S.P.; validation, S.L.

**Funding:** This study was carried out with the support of "Research Program for Agricultural Science & Technology Development (Project No. PJ014207)", National Institute of Agricultural Sciences, Rural Development Administration, Republic of Korea.

**Conflicts of Interest:** The author certifies that he has NO affiliations with or involvement in any organization or entity with any financial interest (such as honoraria; educational grants; participation in speakers' bureaus; membership, employment, consultancies, stock ownership, or other equity interest; and expert testimony or patent-licensing arrangements), or non-financial interest (such as personal or professional relationships, affiliations, knowledge or beliefs) in the subject matter or materials discussed in this manuscript.

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
