# Peer review of "Optimum Method Uploaded Nutrient Solution for Blended Biochar Pellet with Application of Nutrient Releasing Model as Slow Release Fertilizer"

_applsci, doi:10.3390/app9091899_

Round 1

Reviewer 1 Report

Overall, it is a reasonably well done study. My biggest problems are with the writing. There are many fairly obvious English language mistakes. And there are areas where the content itself falls short. For example, there are many possible treatments to test for the biochar-pig manure pellets, but they do not do a good job explaining why they selected their treatments. The discussion could do a better job of showing where their work fits in the existing literature, and perhaps more on the practical importance. 

I also felt that the graphs were of low quality. I think it is a worthy topic and that the authors are competent scientists. With some careful editing I think they will have a worthy paper. 

Author Response

The author thanks to the Reviewer for his valuable comments on this manuscript which has now been adjusted accordingly. The revised portions of the manuscript are in green font.

Reviewer 2 Report

This manuscript is devoted to applying the kinetic model for nutrient release from biochar pellets. The authors investigated the hyperbola type kinetic model for nutrient release and compared various biochar pellets prepared with different treatments. However, there were several points that I could not understand well. I recommend extensive revision before accepting this manuscript. Some of the problems I met during my reviewing are listed as follows. (1) What is the purpose of this study? To develop a new kinetic model for nutrient release from biochar pellet, or, to compare the performance of various biochar pellets prepared with different treatments? The title seems to suggest that the application of the kinetic model is a central topic in this study, but the abstract is written based on the comparison of the various biochar pellets. Does the kinetic modeling give more information than just comparing the released amount of nutrient on each day for the optimization of biochar pellets? What is the information in this study that the authors could not obtain if they do not use the kinetic model? (2) As the authors mentioned in the discussion section, the tendency of NO3-N release can be divided into two phases. Is it appropriate to apply one equation for a different phase of nutrient release? I think that the solid lines in the right graph of Figure 1 can not simulate the nutrient release observed experimentally. (3) I could not understand the meaning of "a," "b," "c," "d," "ab," and "ns" in Table 8.

Author Response

The author thanks to the Reviewer for his valuable comments on this manuscript which has now been adjusted accordingly. The revised portions of the manuscript are in red font.
